

# The effect of using games in teaching conservation

Cedric Kai Wei Tan[1], Jiin Woei Lee[2], Adeline Hii[3], Yen Yi Loo[4], Ahimsa Campos-Arceiz[3,5] and David W. Macdonald[1]

[1] Wildlife Conservation Research Unit, Department of Zoology, University of Oxford, The Recanati-Kaplan Centre, Tubney, Oxfordshire, UK
[2] Graduate School, University of Nottingham–Malaysia Campus, Selangor Darul Ehsan, Selangor, Malaysia
[3] School of Environmental and Geographical Sciences, University of Nottingham–Malaysia Campus, Selangor Darul Ehsan, Selangor, Malaysia
[4] Division of Biology & Conservation Ecology, School of Science & the Environment, Manchester Metropolitan University, Manchester, UK
[5] Mindset Interdisciplinary Centre for Tropical Environmental Studies, University of Nottingham–Malaysia Campus, Selangor Darul Ehsan, Selangor, Malaysia

Corresponding author
Cedric Kai Wei Tan,
cedric.tan@zoo.ox.ac.uk

## ABSTRACT

Games are an increasingly popular approach for conservation teaching. However, we know little about the effectiveness of the games on students' experiences and knowledge acquisition. Many current games are supplemental games (SG) that have no meaningful interaction with the subject matter. We adapted the experiential gaming (EG) model where students were immersed in goal-orientated tasks found in real-life situations, and they tackled questions to complete actions for their main task. Classroom-based games were created for eight different conservation topics for an annual Wildlife Conservation Course and an annual Diploma in International Wildlife Conservation Practice. Data were collected over two cycles, a total sample size of 55 multinational students. We used a combination of repeated-measures design and counterbalanced measures design; each student was subjected at least twice to each of the EG and didactic instruction (DI) treatments, and at least once to the SG approach. We compared students' perception, learning and behavioural responses to the treatments, including measures of student personality types and learning styles as explanatory variables. Findings revealed multiple benefits of the classroom EG compared to the DI approach, such as increased attention retention, increased engagement and added intrinsic motivation. The improved level of intrinsic motivation was mainly facilitated by increased social bonding between participants. Further, we show that this EG approach appeals to a wide range of learning styles and personalities. The performance of SG was generally intermediate between that of EG and DI. We propose EG as a beneficial complement to traditional classroom teaching and current gamified classes for conservation education.

## INTRODUCTION

Conservation education aims at developing lifelong knowledge and skills relevant for conservation action (*Hungerford & Volk, 1990*). As the human population continues to increase, intensifying the demand for natural resources, there is an increasing need for improved education and outreach methods to effect attitude and behavioural change. In recent years, many techniques have emerged to integrate the conservation message with forms of delivery that immerse learners in different perspectives. Some of these delivery methods include hands-on activities, role-play and games (*Jacobson, McDuff & Monroe, 2015*).

When referring to games here, we mean 'immersive, voluntary and enjoyable activities in which a challenging goal is pursued according to agreed-upon rules' (*Kinzie & Joseph, 2008*). Given the potential to enhance engagement and motivation, environmental educators have utilised games to teach and learn (*Hewitt, 1997*; *Bromley, 2000*). Conservation often involves hard choices, compromises and even conflict (*Redpath et al., 2013*), all of which provide rich source material for game creation. By incorporating active learning principles, games can also empower pupils to exercise responsibility for their own lives and for the environment (*Tilbury, 1995*). Thus, games can be a powerful tool for demonstrating and teaching conservation concepts to both children and adults (*Project WILD, 2000*). However, despite their increasing popularity in conservation education (*Bång, Svahn & Gustafsson, 2009*), we are unaware of published empirical evidence on the effectiveness of conservation games. Additionally, when examining the effects of games in education, while some studies have demonstrated benefits (*Klein & Freitag, 1991*; *Wang & Chen, 2010*), others have shown negative effects on motivation and performance (*Kirriemuir & McFarlane, 2004*). This inconsistency could be attributed to a variety of reasons, from the different game forms to the different personalities and learning styles of the students (*Hill et al., 2003*; *Rapeepisarn et al., 2008*; *Lara, 2013*).

Many of the games used in education support the repetition of and hence reproduction of facts (*Boocock, 1966*; *Boocock & Schild, 1968*). The nature of these drill and practise games may, unintentionally and as a result of the absence of real consequences when playing the game, foster a tendency in the players to attempt actions with little reflection on outcomes. As such, players may simply continue experimenting with different actions until their scores improve. Although these games have the potential to increase engagement in conservation teaching, and the fun of the teaching experience, they can involve competition for extrinsic rewards (points, badges, or token movements) for the participants. Further, they do not provide control over, or responsibility for, the outcome (*Room 10 of Hastings Central School, 2013*). Games of this genre also have limited interaction with the subject matter. Henceforth, we term such games as supplemental games (SG), implying that while they are additional (and potentially enjoyable and informative) tools for learning, they do not facilitate learning through the play experience. An example of a supplemental conservation game is the 'Freshwater Board Game' where players move around a board and face ecological events that help them advance or move backwards (*Room 10 of Hastings Central School, 2013*). Players learn about freshwater

ecosystem but they have no control over their actions or movements which are mainly determined by the dice roll.

Another form of gaming, experiential gaming (EG), differs from supplemental gaming in that players learn about the subject matter by participating in or simulating the process and making judgements with consequences (*Kiili, 2005*), for example by managing a population of endangered species. Importantly, EG emphasises immediate feedback from players' actions, giving the players relevant challenges while they work towards a goal (*Kiili, 2005*). This EG model integrates educational theories and game design to facilitate understanding and optimal learning for participants (*Kiili, 2005*). For example, in the board game 'Conservation Crisis,' players manage a wildlife reserve with a species on the verge of extinction (*Gilhead & Milburn, 2016*). They must strive to protect their wildlife and mitigate conservation challenges. Hence, players learn not only about the facts of a case, but also about the challenges faced by conservationists when implementing measures judged necessary to protect wildlife. This type of game emphasises the *application* of knowledge and students learn about the *consequences* their actions and decisions (*Kiili, 2005*). Therefore, they are ideal for teaching dynamic and complex systems with time-delays and feedback on decisions. Conservation biology is framed exactly within such a dynamic and complex systems, and therefore is, as we demonstrate here, fertile ground for the development and use of EGs.

The different game forms might also interact with the learning styles and personality traits of learners to give rise to different perceptions, behaviour and degree of learning from individual learners. Previous studies have demonstrated that an individual's academic performance is an outcome of an interplay between the educational method, learning style and personality attributes (*Chamorro-Premuzic et al., 2008*; *Richardson, Abraham & Bond, 2012*). For example, students with high agreeableness and low neuroticism have been shown to prefer more interactive lessons such as group work and practical teaching (*Chamorro-Premuzic, Furnham & Lewis, 2007*). In another study, achievement motivation was positively correlated with the meaning, reproduction and the application-directed learning style, and negatively correlated with the undirected learning style (*Busato et al., 1998b*). Yet, there have been few studies investigating the interplay between these personal factors and an individual's affinity for the different forms of games (*Hill et al., 2003*; *Rapeepisarn et al., 2008*; *Lara, 2013*). This is pertinent especially in the context of conservation education, which is usually aimed at a large and diverse range of audiences. Our study aims to address this, examining the effectiveness of the different gaming approaches on individuals exhibiting different learning styles as well as personality traits.

Research in this area is needed to establish the extent to which games really can contribute to conservation teaching. Here, we tested the effectiveness of different game forms (SG and EG) and compared these with the approach of didactic instruction when teaching conservation. We quantified perception through the use of questionnaires, learning effectiveness with quizzes and behaviour via observations and video-recording. Additionally, we assessed whether these responses to different lesson types were affected by the students' personality types and learning styles.

**Table 1 Cohort demographics table.**

| Course | Year | Location | Number of students | Gender distribution | Age distribution (years) | Notes on questionnaire on perception | Experimental design (Number of topics) |
|---|---|---|---|---|---|---|---|
| Wildlife Conservation Course | 2015 | University of Nottingham Malaysia Campus | 21 | 10 males; 11 females | 1 (<20); 8 (21–25); 8 (26–30); 2 (31–35); 2 (>35) | In section 1, the statements 'Appreciation of application' and 'Degree of connection with peers' were not in the questionnaire. Section 2 on intrinsic motivation and section 3 on bondedness were not in the questionnaire | Counterbalanced measures design (3); Repeated-measures design (1) |
| Wildlife Conservation Course | 2016 | University of Nottingham Malaysia Campus | 16 | 4 males; 12 females | 8 (21–25); 4 (26–30); 2 (31–35); 2 (>35) | All sections and questions were in the questionnaire | Counterbalanced measures design (3); Repeated-measures design (2) |
| Diploma in International Wildlife Conservation Practice | 2015 | University of Oxford Wildlife Conservation Research Unit | 8 | 4 males; 4 females | 2 (21–25); 5 (26–30); 1 (>35) | In section 1, the statement 'Appreciation of application' with peers' was not in the questionnaire. Section 2 on intrinsic motivation and section 3 on bondedness were not in the questionnaire | Repeated-measures design (3) |
| Diploma in International Wildlife Conservation Practice | 2016 | University of Oxford Wildlife Conservation Research Unit | 8 + 2 | 5 males; 5 females | 2 (21–25); 7 (26–30); 1 (31–35) | All sections and questions were in the questionnaire | Repeated-measures design (5) |
| **Total** | | | 55 | 23 males; 32 females | 1 (<20); 20 (21–25); 24 (26–30); 5 (31–35); 5 (>35) | | |

Note:
Repeated-measures design: a single group of students were taught the first session with one lesson type and then the second session with another lesson type.
Counterbalanced design: class was divided into two groups and both were subjected to a two-session lesson. During the first session, one group underwent one lesson type and the other underwent the other lesson type. In the second session, the treatments were swapped and thus each group experienced both types of lesson in a balanced ordered manner. Details on the experimental design used and content taught for each topic and session are provide in Table 2.

## METHODS

### Participants and course details

Our study was conducted on 55 conservation biologists from two courses: (1) a Wildlife Conservation Course (WCC) organised by Wildlife Conservation Research Unit (WildCRU, University of Oxford) and Nottingham University in Malaysia and; (2) the Recanati-Kaplan Centre Postgraduate Diploma in International Wildlife Conservation Practice in Oxford, United Kingdom (Table 1). Personnel were early career conservation biologists who have been working in conservation organisations for fewer than five years or were doing their graduate studies in conservation biology.

Twenty-one conservation biologists from different countries in Southeast Asia were selected to attend the WCC that took place between 19th and 30th January 2015. In 2016, the same course was held again on 4th to 15th January for 16 other participants. This course was held in the University of Nottingham Malaysia Campus (UNMC). Overall, participants consisted of 14 (37.8%) men and 23 (61.2%) women. One participant was 16–20 years-old, 16 participants were 21–25 years-old, 12 were 26–30 years-old, four were 31–35 years-old and four were >35 years-old. The course comprised topics ranging from concepts in conservation biology, to transferrable and practical skills. This study was approved by the University of Nottingham Malaysia Campus Science & Engineering Research Ethics Committee (Ref No. JL051115 & CT011114) and participants involved willingly signed an agreement form prior to the lessons.

Another set of participants comprised 16 international students (eight each for year 2015 and 2016) attending the seven-month Recanati-Kaplan Centre Postgraduate Diploma in International Wildlife Conservation Practice directed by WildCRU, University of Oxford. Four participants were 21–25 years-old, 10 were 26–30 years-old, one was 31–35 years-old and one was >35 years-old. There were four men and four women each year. The taught Diploma covered a range of subjects in conservation biology. In three to four topics of the Diploma, the participants were subjected to different lesson types and data on these participants collected for this study. In year 2016, two PhD students from the WildCRU attended the lesson for two topics and participated in the games and DI classes. They were included in the study to add to the sample size. This study was approved by the University of Oxford Central University Research Ethics Committee (Ref No: R42538) (*Oxford University, 2017*) and participants involved willingly signed an agreement form prior to the lessons. Table 1 summarises the cohort demographics for each course for separate years.

## Personality and learning style questionnaires

Prior to the start of the course, questionnaires were administered to participants to assess their personality traits and learning styles. The Five Personality Factor test, 5PFT″ (70 statements) is the first published personality questionnaire specifically created to measure the personality traits now known as the big five (*Elshout & Akkerman, 1975*): agreeableness, conscientiousness, extroversion, neuroticism and openness. The big five model is one of the most established and recognised approaches in measuring individual differences in personality (*Costa & McCrae, 1992*).

A recent review identified 71 different learning style models (*Coffield et al., 2004*). It can thus be difficult researchers to select an appropriate learning style assessment (*Cassidy, 2004*). We used three criteria to select our learning style model: (1) whether the purpose of the model matches the aim of the study, (2) how extensively the model has been used in past studies and (3) the validity and reliability of the associated instrument (*De Bello, 1990*). Thus, we used the inventory of learning styles (ILS) (120 statements) (*Vermunt, 1994*). This model measured different aspects of processing strategies, regulation strategies, learning orientations and mental models of learning (*Vermunt, 1994*). Vermunt's model of learning has been used widely within the higher education community (*Busato et al., 1998a*;

*Boyle, Duffy & Dunleavy, 2003*) and has been used to examine the relationship between learning style, the big five personality traits and academic performance (*Quinn et al., 2017*; *Busato et al., 1998b*). Students answered the inventory and scores were subsequently calculated for each component of Vermunt's four learning styles: undirected, reproduction-directed, application-directed and meaning-directed.

For both questionnaires, statements within each component overlapped considerably and were a mix of positive and negative, increasing the validity and reliability of the assessment (*Mount, Barrick & Strauss, 1994*; *Vermunt, 1998*). Details on the definition of each category of personality and learning style are provided in Method S1.

## Lesson types

We employed three lesson types: DI, SG) or EG for our classes. The guidelines used to create the games are explained in Method S1. Each topic was divided into two sessions that had complementary content. For example, the first session on behavioural ecology was about how to conduct behavioural ecological research and second session was on the link between behavioural ecology and conservation biology. Table 2 provides information on the content taught for each session of each topic. For one topic, either one (SG or EG) or two games (SG and EG) were developed. This allowed for comparisons of the students' responses to SG *vs* DI, EG *vs* DI or SG *vs* EG. For the Diploma 2016 students, the R analysis topic had three sessions each with a different lesson type (SG or EG or DI) to examine the response to three lesson types within one topic.

All topics were taught by the same tutor, irrespective of lesson type. This minimised any potential confounds when using different tutors. Regardless of the lesson type, the content and concepts taught for each session were the same. To ensure this, we used the same lesson slides for all lesson types, only adding slides to introduce the game in SG and EG. In all lesson types, students were grouped in teams of two to four people to discuss questions asked by the tutor and to work on the given tasks. This standardises the number of opportunities to answer questions in all lesson types.

## Experimental design

We adopted two experimental designs to examine the differences between lesson types: repeated-measures design or counterbalanced measures design. In a repeated-measures design, a single group of students was taught the first session with one lesson type (DI, SG or EG) and then the second session with another lesson type (Fig. 1A) (*Von Ende, 2001*; *Lawal, 2014*). In a counterbalanced measures design, the class was divided into two groups and both were taught a two-session lesson. During the first session, one group was taught using one lesson type and the other was taught using the other lesson type. In the second session, the treatments were swapped and thus each group experienced both types of lesson in a balanced ordered manner (Fig. 1B) (*Shuttleworth, 2009*).

The design used was dependent on the number of students and the time available. The games required a minimum of eight players (for four teams with at least two persons each). Therefore, the repeated-measures design was used for the Diploma course as there were only eight students (Tables 1 and 2). We recognise that the repeated-measures design

**Table 2** Topics taught during the Wildlife Conservation Course (WCC) and Diploma in International Wildlife Conservation Practice and the experimental design used.

| Topic | Lesson type (mean duration ± SD in min) | Content of each session | Task in game | Experimental design | Course |
|---|---|---|---|---|---|
| Behavioural ecology | EG (125 ± 19 min) and DI (62 ± 17 min) | First: Behavioural ecology research<br>Second: the link between behavioural ecology and conservation biology | Teams were species of mammals interlinked via a food web and had to behave like the species to reproduce, feed or avoid being predated by other teams | Counterbalanced<br><br>Repeated-measures (EG then DI) | WCC 2015 and 2016<br><br>Diploma 2016 |
| Conservation genetics | EG (100 min) and DI (45 min) | First: The effects of the environment on genetic processes<br>Second: Role of genetics in conservation biology | As conservation geneticists, teams had to manage a population of coloured casino chips by betting on their answers. Different colours represented different phenotypes. The goal is to avoid stochastic and genetic events (e.g. genetic drift) and attain a population as heterogeneous as possible | Repeated-measures (DI then EG) | Diploma 2015 |
| Human-wildlife conflict | EG (159 ± 31 min) and DI (63 ± 25 min) | First: Preservation of cultural traditions, canned lion hunting<br>Second: Coexistence or separation, compensation schemes | Assuming the roles of government, conservation biologists, rural population or urban population, teams are given the task of managing a forest where human-wildlife conflict is prevalent | Counterbalanced | WCC 2015 and 2016 |
| Capture-mark-recapture | EG (122 ± 20 min) and DI (110 ± 14 min) | First: Designing camera trap studies<br>Second: Analysing camera trap data | Teams were conservation biologists tasked to buy different models of camera traps, deploy them in a forest board and subsequently analyse the collected data to reveal the density of clouded leopards | Repeated-measures (EG then DI) | Diploma 2015 and 2016, WCC 2016 |
| Vegetation statistical analysis | SG (150 min) and DI (60 min) | First: Comparing tree species and density between two habitat types<br>Second: Analysing Global Forest Cover map data | Teams aimed to either obtain the most number of chocolates (SG) or prevent the continuous vegetation from being disconnected (EG) | Repeated-measures (SG then DI) | Diploma 2016 |
| Spatial-temporal patterns | EG (132 ± 21 min) and DI (90 ± 10 min) | First: Analysing spatial patterns of animals<br>Second: Analysing temporal patterns of animals | Teams were species of mammals attempting to either avoid (as prey) or overlap (as predator) the activity of other teams | Counterbalanced<br><br>Repeated-measures (DI then EG) | WCC 2016<br><br>Diploma 2016 |
| Population viability analysis | SG (90 ± 18 min) and DI (48 ± 4 min) | First: Dynamics of small populations<br>Second: Analysis of population viability | Teams were to obtain gummies by answering the questions correctly | Counterbalanced | WCC 2015 |

(Continued)

| Topic | Lesson type (mean duration ± SD in min) | Content of each session | Task in game | Experimental design | Course |
|---|---|---|---|---|---|
| R analysis | SG (110 ± 30 min), DI (80 min) and EG (120 ± 16 min) | First: Statistical concepts (SG) Second: Hypothesis testing (DI) Third: Experimental design (EG) | Using the software R, teams were answer questions in order to advance forward on a game board (SG) or to obtain tokens for designing an experiment (EG) | Repeated-measures (SG then EG) Repeated-measures (SG then DI then EG) | Diploma 2015, WCC 2015, 2016 Diploma 2016 |

**Note:**
Details on the experiential gaming (EG) task or supplemental game (SG) task assigned to the teams are provided. Repeated-measures design: a single group of students were taught the first session with one lesson type and then the second session with another lesson type. Counterbalanced design: class was divided into two groups and both were subjected to a two-session lesson. During the first session, one group underwent one lesson type and the other underwent the other lesson type. In the second session, the treatments were swapped and thus each group experienced both types of lesson in a balanced ordered manner.

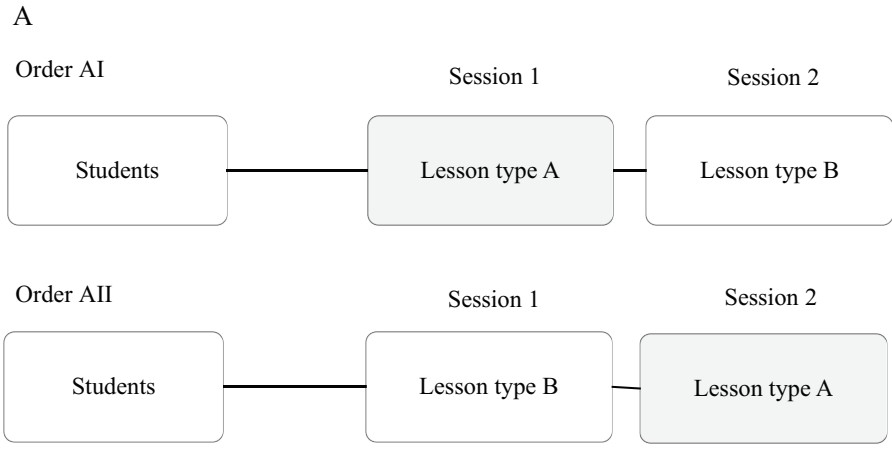

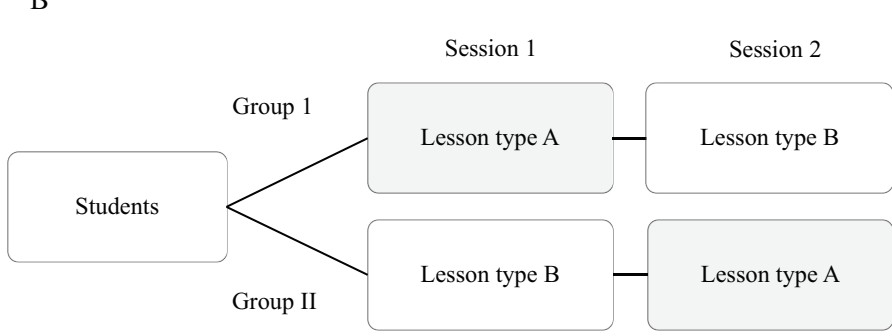

**Figure 1 Experimental design.** (A) Repeated-measures design. Each student experienced two lesson types (DI, EG, SG) in either the order AI or the order AII. Where possible, we alternated the order (AI *vs* AII) of the lesson type for different topics and for different groups of students (please see Table 2 for details). (B) Counterbalanced measures design. Students were divided into two groups and one group is treated with lesson type A, followed by lesson type B, and the other is tested with lesson type B followed by lesson type A.

has a limitation in which students' response to the second session depended on the lesson type of the first session, i.e. order effect (*Von Ende, 2001*; *Lawal, 2014*). Therefore, when possible, for the same cohort of students, we alternated the order of the lesson type for different topics to minimise potential treatment order effects (Table 2). Because of the limitations in time and resources (one tutor), not all students were exposed to all topics. However, we were able to ensure that all students except for the two PhD students were exposed at least twice to EG, twice to DI and once to SG to allow for within-individual comparisons. The two PhD students were exposed twice to EG and once to DI lesson types.

The counterbalanced measures approach utilised twice the amount of time and half the number of students each time as compared to the repeated-measures design. Therefore, the counterbalanced measures design would be limited to the course with more time and with a larger group of students, i.e. the WCC. This design reduces the chances of the treatment order influencing the results (*Shuttleworth, 2009*). For example, the behaviours of the subjects might be affected by the order in which treatments were given, due to fatigue or other reasons. However, due to time limitations, we also used the repeated-measures design for a few topics of the WCC (one topic in year 2015, two topics in 2016; Table 2). Details of the experimental design and order utilised for the different groups of students are provided in Table 2.

## Assessment

Participants were assessed in three ways: (1) perception: via questionnaires administered to participants, collecting data of individual perceptions of each lesson (Supplemental Information 1); (2) behaviour: via observational surveys and video recording during the classroom sessions and; (3) learning: via quizzes pre and post-session to examine the knowledge acquired by each individual. These modes of assessment overlapped considerably and hence served to cross-validate the students' responses to the different lesson types (*Dirks, Wenderoth & Withers, 2014*).

### (1) Perception

The questionnaire designed to test students' perceptions was divided into three sections. In the first section, students were asked to compare the lesson to that of a traditional teaching method experienced previously (not during the course) and rate each statement (e.g. amount of content taught) according to whether they strongly disagreed (1 point), disagreed (2 points), no difference (3 points), agreed (4 points) and strongly agreed (5 points) (Supplemental Information 1). These statements reflected the wide range of perceptions held by the learners about their behaviour and learning (*Black & Wiliam, 1998*). In 2014, this first section of the questionnaire was tested on 20 undergraduates of the University of Oxford as well as on eight students of the Diploma course. Subsequently, it was edited by three of our co-authors prior to the conduct of this study (year 2015 onwards). This section was modified slightly after the earlier courses in 2015. The statement 'Degree of connection with peers' was added to the questionnaire after the first course (WCC 2015) and the statement 'Appreciation of application' was added for students of 2016. Sections 2 and 3 of the questionnaire were only administered

to students of 2016. These extra statements and sections were added after considering the feedback from students in earlier courses in 2015: Students wrote in their feedback that these were the other beneficial aspects of game lessons. The sample sizes for these additional statements of the questionnaire are thus smaller than that of the other statements.

The second section of the questionnaire is a shortened version of the intrinsic motivation inventory (*Ryan, Koestner & Deci, 1991*), which focuses on the degree of intrinsic motivation and its predictors: perceived choice, perceived competence, pressure/ tension and social bonding (see Method S1 for details). These four subscales each had five statements which overlapped considerably and were a mix of positive and negative, increasing the validity of the assessment (*McAuley, Duncan & Tammen, 1989*; *Tsigilis & Theodosiou, 2003*). The third section was aimed at measuring interpersonal interactions between students within teams, which is a positive predictor of intrinsic motivation (*Ryan, Koestner & Deci, 1991*). It was based on the Inclusion of Other in Self Scale (*Aron, Aron & Smollan, 1992*): 'Please circle the picture that best describes your relationship to the other participants in your group' (Supplemental Information File (Questionnaire)).

### (2) Behaviour

We noted the frequency of joyful behaviour (smiling, laughing, clapping, cheering), distracted behaviour (yawning, falling asleep expressions, looking at phones), asking questions and answering questions for each individual (Supplemental Information File (behavioural observation sheet)). We only recorded asked questions that were relevant to the topic, rather than any questions relating to the game or administration. Behaviours were selected based on three criteria. First, at similar classes prior to the study, a catalogue of the typical students' behaviour was created and refined to answer our research questions (Supplemental Information File (Behavioural observation sheet)). Second, these behaviours complemented the statements used in the perception questionnaire, for example, the 'degree of encouragement to ask questions' and hence served to verify the perception response. Third, several of these behaviours have been used in previous studies on undergraduates to understand student participation (asking and answering questions, *Fritschner, 2000*; *Mustapha, Rahman & Yunus, 2010*) and distraction (phone-using and sleeping, *McCoy, 2016*; *Mustapha, Rahman & Yunus, 2010*). Particular behaviours were biased to certain lesson types, e.g. joyful behaviour is expected to be higher when engaging in a game than when attending a DI lesson. Nevertheless, we were also interested in examining whether these behaviours were different between the two game forms. For example, when a SG is distracting, it might elicit fewer joyful behaviours than a EG lesson. Due to logistical constraints, we did not record behaviour for the R analysis lessons in the WCC 2015. Behaviours that occurred for a duration of more than a minute were considered as multiple counts if there was at least a 10 s interval between the behaviours (*Mahar et al., 2006*). To standardise the number of opportunities of asking question, during the DI lessons, questions used in the game lessons were asked verbally and the tutor waited for answers from the participants.

Behavioural data were collected via observer note-taking and subsequently verified with video records (*Mustapha, Rahman & Yunus, 2010*). Observers were trained prior to lessons to minimise inter- and intra-observer bias. They were trained by JWL at lessons not analysed in this study. Two observers and one observer were employed for observations during the WCC and Diploma programme, respectively. Since courses were carried out in different countries, different observers were involved. Therefore, after the courses, another observer who was not involved in direct note-taking made behavioural recordings from the videos filmed during class. Our inter-observer correlation coefficients were 0.71, 0.78, 0.72 and 0.80 for Diploma 2015, Diploma 2016, WCC 2015 and WCC 2016, respectively and thus above the recommended criterion of >70% or greater (*Jonsson & Svingby, 2007*). To ensure consistency, we used the behavioural data collected by the last observer for our analyses.

### (3) Learning

Quizzes consisted of questions asking facts, meaning or application of topic to a real-world situation. Therefore, a question could be categorised as reproduction-, meaning- or application-based question type, guided by the revised Bloom's taxonomy (*Krathwohl, 2002*) and complementary to the learning styles quantified by Vermunt's ILS. The Cognition Process Dimension of the revised Bloom's taxonomy consist of six levels that are defined as Remember, Understand, Apply, Analyze, Evaluate and Create. We only constructed questions that tested the lowest three levels (Remember, Understand, Apply) as the higher three levels required more complex testing procedures. Our questions on reproduction (remembering) required students to 'retrieve, recognise and recall relevant knowledge from memory' (*Krathwohl, 2002*); for example, 'what is inbreeding depression? Our meaning-based questions required learners to 'construct meaning from messages through interpreting, exemplifying, classifying, summarising, inferring, comparing and explaining' (*Krathwohl, 2002*). An example is 'What is the difference between raster and vector? Finally, our application-based questions required students to 'carry out or use a procedure in a given situation' (*Krathwohl, 2002*). An example of a question is 'How do we apply the stratified random sampling method when interviewing villagers? The questions were in a multiple-choice format with four options, only one of which was right. There were 10 questions for each quiz, comprising a mixture of question types. The same questions were administered before and after each session. In 2016, a third quiz with the same questions was conducted one week after the lesson to quantify retention of knowledge after a longer period of time (as compared to immediately post-lesson). We determined the difficulty level and discriminatory level of each quiz by calculating the mean and standard deviation of correctness. If a question was more difficult, the mean 'correctness' would be closer to zero and if it were easier, the mean 'correctness' would be closer to one. The larger the standard deviation of correctness, the larger the variation in correctness among students and therefore the more discriminatory the question was.

### Statistical analysis

All analyses were conducted using R 3.0.2 and with mixed models due to the repeated measures obtained (*Ellison, 2017*; *Tango, 2017*). To examine the effects of lesson type

(DI, SG or EG) on students' perceptions, we used a multinomial mixed model with level of agreement as the response, lesson type as the fixed factor, and student and topic as random factors. Student was entered as a random factor as there were multiple questionnaires completed by each student (e.g. different sessions but same topic). Topic was entered as a random factor to account for any potential differences among topics. The covariates included were year, course (WCC or Diploma), age, gender, teaching experience (yes or no), formal teaching training (yes or no). We used one model for each statement of the perception questionnaire (e.g. Encouragement to ask questions).

We analysed the variation in each behaviour (joyful, distracted, question-asking and question-answering) with two analyses: (i) a generalised linear mixed model (GLMM) with a binomial error distribution, the 'probability of performing the behaviour' as a response variable, and (ii) a linear mixed model with a normal error distribution and the 'rate of behaviour per hour' as the response variable, omitting zeros.

To examine whether learning effectiveness differed with lesson type for the three topics, we entered in a GLMM, with binomial error distribution, correctness as a response (1 = correct; 0 = wrong), lesson type and period (before or after or one-week post lesson (for 2016 only)) and their interaction as fixed factors, topic and student as random variables. Details on checking for assumption violations, and fixed factors and random factors entered for each analysis are provided in Method S1.

To account for variation in question difficulty and discriminatory level, we also conducted GLMM analyses with the standardised 'correctness' as the response. That is, each individual response (0 or 1) was standardised to the mean 'correctness' obtained by all individuals answering that question and then we divide this difference by the corresponding standard deviation of the 'correctness.' Therefore, if a question were more difficult, the mean 'correctness' would be closer to zero and if it were easier, the mean 'correctness' would be closer to one.

## RESULTS

### Perception

Overall, students perceived EG as the most beneficial and SG as intermediately beneficial among the lesson types. Students scored EG and SG significantly higher than DI in the following questionnaire statements: 'remembrance of content,' 'motivation to learn after lesson,' 'broadens perspective,' 'nurtures creativity,' 'challenging,' 'attention retention,' 'engagement with tutor' and 'engagement with student' (Figs. 2A–2C; Tables S1 and S2). For the statement 'learning from peers' only EG was scored significantly higher than DI, SG attained an intermediate score between EG and DI (Figs. 2A–2C; Table S1). EG was scored significantly higher than SG and DI in the questionnaire statements 'connection with peers' and 'appreciation of application' (Figs. 2A and 2C). SG was scored significantly higher than DI in the questionnaire statement 'better understanding.' Lastly, for statements 'amount of content' and 'encouragement to ask questions,' no significant differences were found among lesson types (Figs. 2A and 2C; Tables S1 and S2).

Intrinsic motivation measured only for 2016 students revealed significantly higher scores of 'interest/enjoyment' for EG than for DI (SG attained an intermediate score;

A

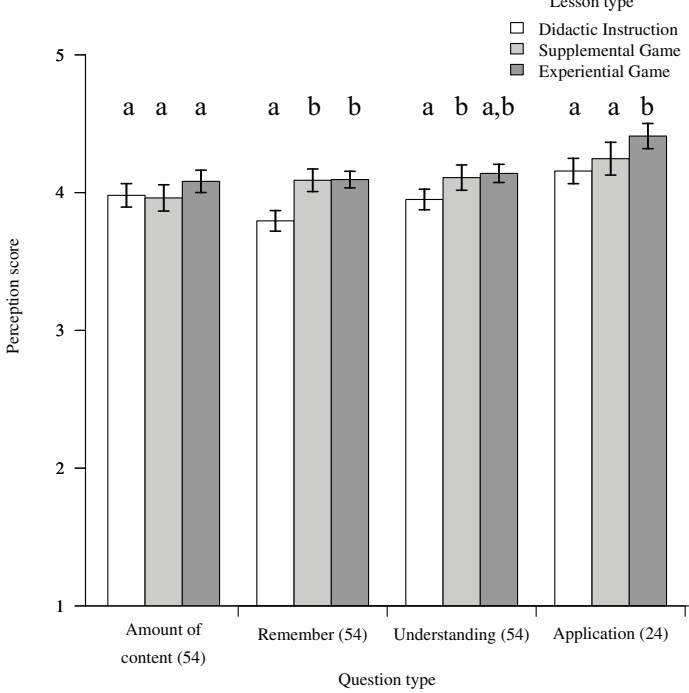

B

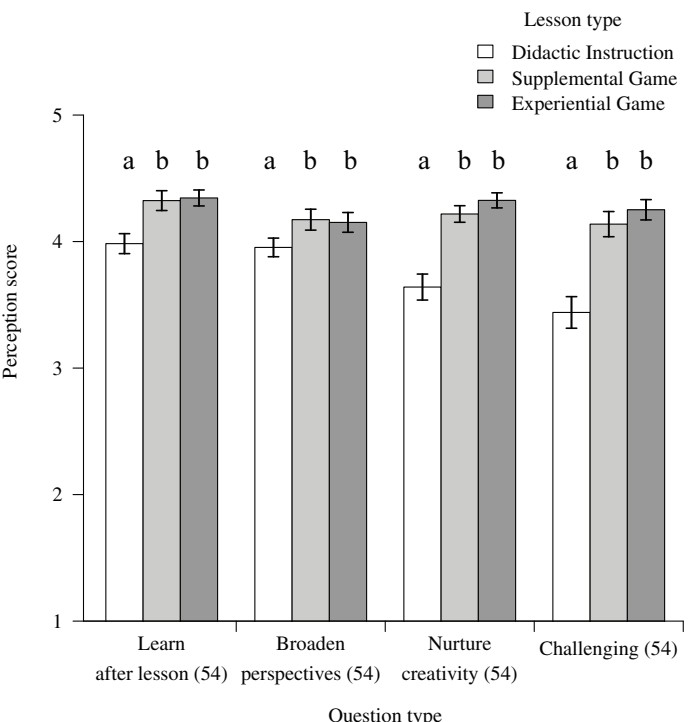

C

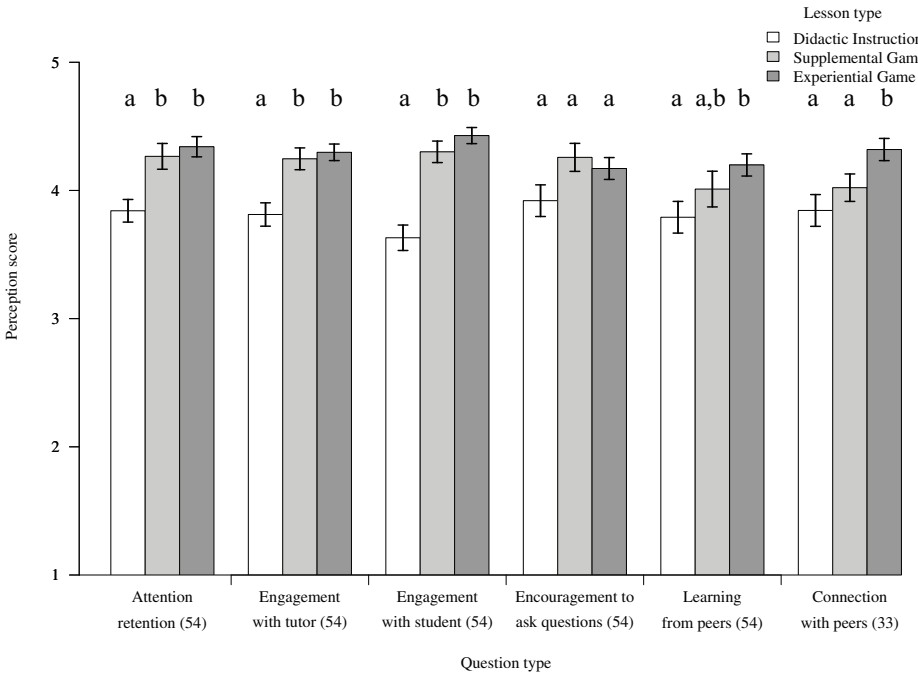

D

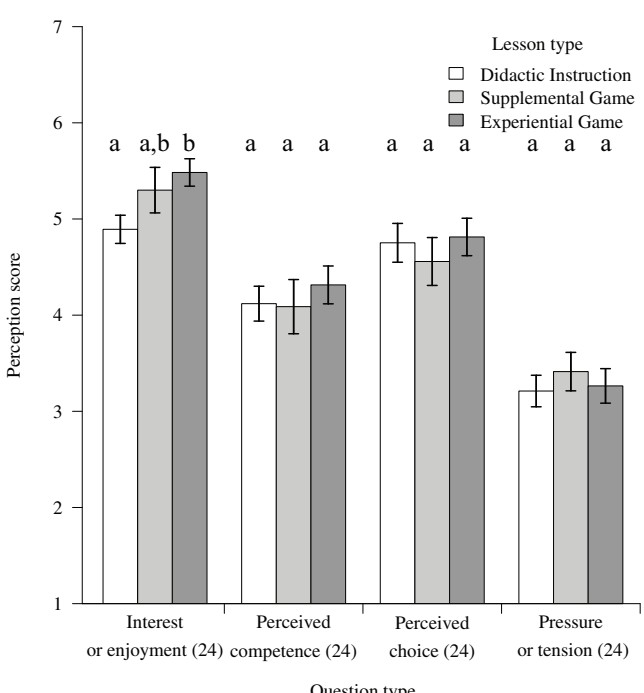

E

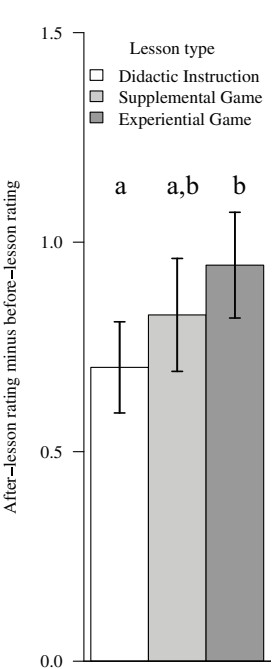

**Bondedness**

**Figure 2  Perception.** Students' perception of the different lesson types as analysed with ordinal logistic models. Students rated their bondedness with their team peers before and after lessons and the *y*-axis shows the after-lesson relationship rating minus that of before-lesson. For each lesson type, the mean perception scores of students are shown. Error bars denote standard errors of sample size (i.e. number of student). Sample sizes are indicated in brackets on the *x*-axis, sample sizes vary because a few of the questions in the questionnaire were altered between years. Different letters above bars denote significant differences, details of results are shown in Tables S1 and S2. (A) Knowledge acquisition parameters; (B) Development parameters; (C) Class dynamics parameters; (D) Intrinsic motivation parameters, 'interest or enjoyment' is considered the self-report measure of intrinsic motivation, 'perceived competence,' 'perceived choice' are positive predictors of intrinsic motivation while 'pressure or tension' is a negative predictor of intrinsic motivation. (E) Bondedness rating, considered a positive predictor of the intrinsic motivation.                    

Fig. 2D). This self-report measure of intrinsic motivation was not explained by 'perceived competence,' 'perceived choice' nor 'pressure or tension' which are predictors of intrinsic motivation (Fig. 2D). Bondedness rating with peers within the team was significantly higher for EG than for DI (Fig. 2E).

When examining whether the overall average rating was affected by personality and learning styles, all except 'Undirected learning style' had non-significant interactions with lesson type on overall rating (Undirected learning score*lesson type: $\chi^2_1 = 7.29$, $p = 0.026$; Table S1). The correlation between overall rating and 'Undirected learning' scores was negligible for EG, and positive for DI and SG (Fig. S1).

## Behaviour

Similar to the results on perception, students' overall behavioural records were most positive in the EG among the three lesson types. There was a significant difference in the probability of 'asking question,' 'joyful behaviour' and 'distraction' occurring among lesson types. Students displayed a higher probability of question-asking for EG lessons

compared to DI and SG lessons, higher probability of joyful behaviour in EG and SG lessons than in DI lessons and higher probability of distracted behaviour in DI than in SG and EG lessons (Fig. 3A; Tables S3 and S4). The frequency of 'joyful' behaviour per unit time was significantly higher in EG compared to SG and DI lessons and the frequency of 'distracted' behaviour per unit time was significantly lower in EG compared to DI lessons (Fig. 3B; Tables S3 and S4).

### Learning

Questions varied in their mean correctness and thus difficulty level (Table S5). They also varied in their standard deviation of correctness and hence level of discrimination (Table S5). On average, questions were sufficiently difficult (mean correctness across lessons ranged from 0.35 to 0.55) and were able to discriminate different degrees of learning (standard deviation of correctness across lessons ranged from 0.144 to 0.244) (Table S5).

Overall, there was no significant difference in students' learning among the lesson types. Students scored higher in quizzes after the lesson than before the lesson. One-week post lesson quiz scores varied (only 2016 students), depending on lesson type. The average one-week post lesson scores were similar to that of the after-lesson quiz.

When analysing all quiz questions, there was no significant difference in learning effectiveness among lesson types (DI *vs* SG *vs* EG), as shown by the lack of interaction between lesson type and period (before or after) on the proportion of quiz answers being correct ($\chi^2_1 = 5.18$, $p = 0.075$; Fig. 4A). This lack of significance of lesson type was consistent across both years (Figs. 4B and 4C; Table S6). However, when examining only reproduction-type questions, there was a significant interaction between lesson type and period on the proportion of quiz answers being correct ($\chi^2_1 = 30.0$, $p < 0.001$; Table S6): the proportional increase in quiz score was higher for DI and EG than for SG (Fig. S2A). This effect was largely attributed to the quiz performance of 2015 students (Figs. S2Ai and S2Aii). When analysing meaning-directed and application-directed question scores separately, the non-significant interaction between lesson type and period indicate that students performed equally well in these questions despite the lesson type (Figs. S2B and S2C; Table S6).

When analysing the variation in standardised score, the results were similar. Among all question types, there was a lack of interaction between lesson type and period (before or after) on the proportion of quiz answers being correct (Table S7). Also, when examining only reproduction-type questions, there was a significant interaction between lesson type and period on the proportion of quiz answers being correct for 2015 students ($\chi^2_1 = 6.23$, $p = 0.044$; Table S7).

## DISCUSSION

We used EG to teach conservation and our results demonstrated multiple benefits of this approach. Based on our behavioural observations and perception data, EGs performed better than DI in retaining the students' attention, increasing their interactions with other students and the tutor, improving intrinsic motivation via social bonding and

A

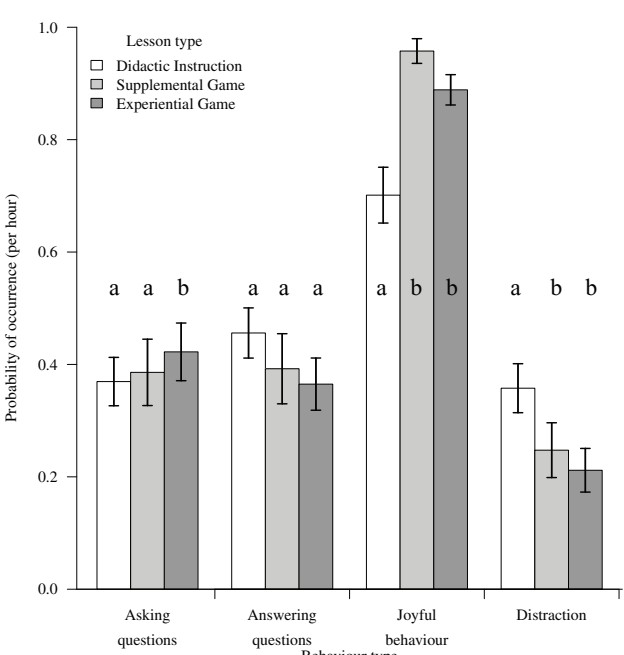

B

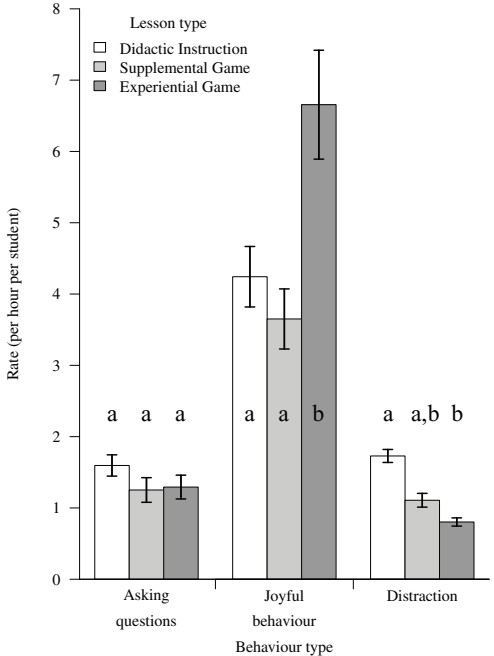

**Figure 3 Behaviour.** Students' behavioural responses to the different lesson types. The probability of behaviour was analysed with a generalised linear mixed model with binomial error distribution and frequency of occurrence was analysed with a generalised linear mixed model with Poisson error distribution. Error bars denote standard errors of sample size ($n = 55$). Different letters above bars denote significant differences, details of results are shown in Tables S3 and S4. (A) Probability of behaviour occurrence; (B) Frequency of occurrence.

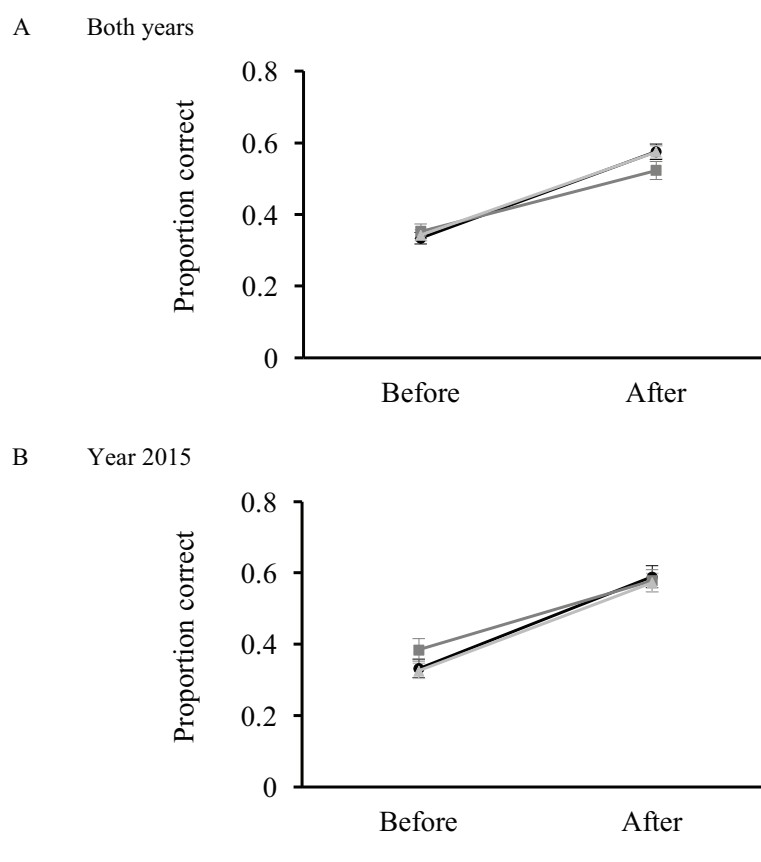

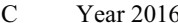

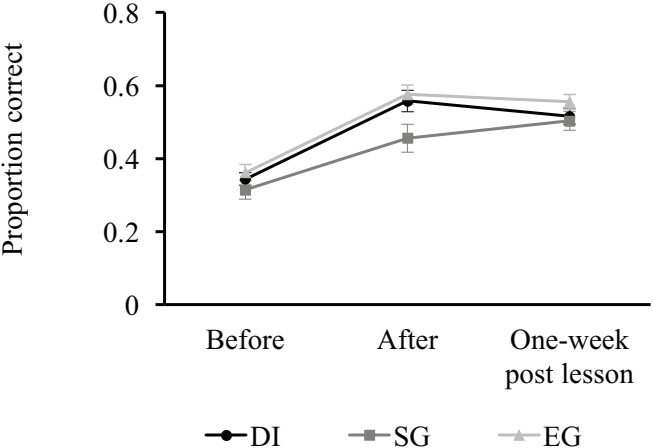

**Figure 4 Learning.** Learning as measured by proportion of quiz questions correct before, after or one-week post lesson. Questions were the same before, after or one-week post lesson and varied in their difficulty and discrimination levels (Table S5). The $y$-axis shows the proportion of questions correct and this was analysed with generalised linear mixed models with binomial error distribution. Error bars denote standard errors of sample size ($n_{both \; years}$ = 55; $n_{year}$ 2015 = 30; year 2016 = 25). Details of results are shown in Table S5. DI, Didactic Instruction; SG, Supplemental Game; EG, Experiential Game. Different lines denote different lesson types. (A) Both years; (B) Year 2015; (C) Year 2016.

motivating students to learn more after the lesson. Additionally, these positive effects of EGs were mostly consistent across personalities and learning styles. EG was as effective as DI in learning effectiveness. SGs performed better than DI in a few perception parameters (e.g. motivation to learn more after lesson) and was mostly on par with DI. SG fared worse than DI in reproduction-type quiz scores.

A student-focused and flexible environment for learning like EG, could develop intrinsic motivation (*Entwistle, McCune & Hounsell, 2002*) and improve the motivation to study and reach success (*Diseth & Martinsen, 2003*). Our study supports this idea: EG fared better than DI in students' intrinsic motivation and the effect of SG was intermediate between that of EG and DI. Extrinsic rewards such as points and leaderboards used in SG can decrease perceived competence and choice and undermine intrinsic motivation (*Deci, Cascio & Krusell, 1975*; *Deci, Koestner & Ryan, 2001*). Although we found non-significant differences in perceived competence and choice between SG and EG, there was a trend of SG attaining lower scores than EG in both these parameters. Increased intrinsic motivation may in turn encourage students to adopt a deeper approach towards learning, which is associated with intentions to understand rather than to reproduce the material (*Fransson, 1977*; *Vansteenkiste, Lens & Deci, 2006*). Moreover, students who prefer teaching methods that are interactive (*Chamorro-Premuzic, Furnham & Lewis, 2007*), or that facilitate understanding of relevance (*Entwistle & Tait, 1990*), are more inclined to use a deep approach. Therefore, we can capitalise on the positive effects of EGs: increased intrinsic motivation and interaction to encourage a deeper approach to learning. Additionally, EGs are especially appropriate for practical conservation like population management and conflict resolution: the understanding of population dynamics and factors affecting conflict respectively would be necessary to the formulation of appropriate decisions.

Experiential games are particularly suited to conservation teaching for many reasons. Commercially available EGs such as task-based simulations (e.g. The Sims (*Electronic Arts, 2009*)) give players a goal or a choice of multiple aims, and players must act consistently in character to achieve it. The benefit of this open-ended concept is that participants can experiment and fail gently. Similar educational EGs such as a jigsaw puzzle to teach Photoshop skills (*Dong et al., 2012*) and calibration games to teach calibration (*Flatla et al., 2011*) work on the same principles of task-based simulation and role-play. They allow for users to experience the process of reaching a set target whilst learning. Indeed, all of these studies demonstrated positive effects in terms of students' perception (*Flatla et al., 2011*; *Smith & Baker, 2011*; *Dong et al., 2012*), and behaviour (*Halan, Rossen & Cendan, 2010*; *Dong et al., 2012*), similar to our findings. The potential role of digital EGs in conservation education has also been recognised for at least 10 years (*Brewer, 2003*). However, research on the effects of using EGs (or games in general) on teaching conservation is lacking. Moreover, seldom are these games designed specifically for the classroom, that is, they are usually computer-based games. This in turn requires a software developer and may also lack real-time interactive feedback from the teacher/peers. Our study demonstrated that the elevated level of intrinsic motivation when playing EGs (relative to DI lessons) was attributed to increased social bonding with peers. This

suggests that EGs might be more beneficial when played in the classroom than on the computer.

Experiential games create opportunities for experiential learning, which is considered more effective than didactic teaching (*Garris, Ahlers & Driskell, 2002*). Experiential learning is student-centred and allows for constant feedback from the students to the tutor. A DI lesson focuses on one-directional fact learning in which meaning or application of the knowledge content may not be effectively assimilated by students. However, we found that EG is at least as good as DI in improving learning, be it reproduction, meaning or application. This is in contrast with other studies that have demonstrated positive effects of EG on learning (*Flatla et al., 2011*; *Smith & Baker, 2011*; *Dong et al., 2012*). In these studies, learning was assessed in terms of whether learners were able to perform the task after the lesson (e.g. using a software). Instead, we assessed learning with multiple-choice questions which were focused on testing whether students assimilated the content and concepts taught. Future works could evaluate how well certain tasks were performed by students.

Our results demonstrated that students underperformed in reproduction-based quiz questions in SG classes relative to EG and DI classes. Because the challenges and actions in a SG do not incorporate meaningful material on the topic, the SG might be considered a distraction from the learning content and jeopardise the recall capacity of the students. Similarly, another study proposes that the controlling game mechanics of SG can decrease intrinsic motivation and hence reduce exam scores (*Hanus & Fox, 2015*).

Two results on the effects of personality and learning styles on the perception of games are noteworthy. We detected the lack of interaction between personality scores and lesson type on overall questionnaire rating. This could be due to the lack of sensitivity of our experimental design and recording tools (e.g. questionnaire) to detect any differences in perception among personality types. Alternatively, it could suggest that the preference of EG over DI might be universal across different personality types. These results are contrary to a study showing that agreeableness and neuroticism were positively and negatively associated (respectively) with preference for interactive lessons such as group work (*Chamorro-Premuzic, Furnham & Lewis, 2007*). Perhaps the EG approach is both interactive and fun, which might appeal to people with low agreeableness (cynical of the world around them) and high neuroticism (tendency to experience negative emotions very intensely). The undirected learning approach is characterised by a lack of discipline and of interest (*Tait & Entwistle, 1996*), and has been shown to be associated with poor academic performance (*Boyle, Duffy & Dunleavy, 2003*; *Kimatian et al., 2017*). Hence, for students with low undirected scores, the preference for EG over SG and DI was not unexpected. SG could be perceived as spontaneous and disorganised and DI could be seen as dry, in contrast to EG where the game is relevant to the topic and students are central to the learning process. It is important to note that the learning style inventory used in this study may not be reflective of the more recent learning style inventories being used, such as the Kolb's experiential learning style inventory (*Tsingos, Bosnic-Anticevich & Smith, 2015*; *Özyurt & Özyurt, 2015*). Kolb's experiential learning style inventory focuses on how knowledge is created through experience (*Kolb, 1984*) and

might therefore be more suitable for our study on EGs. Nevertheless, despite the popularity of evaluating learning styles to optimise education, there is little evidence to suggest that matching activities to one's learning styles improves learning (*Pashler et al., 2009*; *An & Carr, 2017*).

The EG educational approach does not come without caveats. Creating a EG relevant to the topic and obtaining its various game components require more time than when preparing a DI lesson. Additionally, the duration of an EG lesson is on average twice that of a typical DI classroom. We suggest ways of minimising time expenditure in preparing and teaching EG lessons. First, give students a time limit for answering each question (usually 1 min), this would make the lesson time-regulated. Second, tutors can focus on teaching important concepts during class and then provide supplementary material for the students to learn more after the class. Nevertheless, when time is limited and when there is large volume of content to be covered (e.g. as dictated by a syllabus), the approach of EG or any form of games might not be suitable. Further, in developing countries where students might not have access to comprehensive libraries or online access to academic material, the content would have to be provided and explained by the teacher face-to-face.

We are mindful of potential biases in this study. First, the tutor's self-bias for or against a particular lesson type might affect the students' response to that lesson type. We took precautions to minimise this bias by providing a similar number of opportunities to engage with the tutor in all lesson types (i.e. ask or answer questions). The use of a single tutor had the advantage of avoiding any potential confounds between different tutors. It would nevertheless be interesting to investigate the effect of using multiple tutors, each teaching all lesson types. Second, there might be a selection bias in the students used in this study, i.e. students might have been selected based on whether the game approach would appeal to them. This bias was also minimised. The WCC applicants submitted their curriculum vitae and personal statements. Three persons including the tutor graded the applications independently and their scores were averaged to make the final decision. Students in the Diploma programme were selected by an interview panel not including the tutor. Third, the students' self-report of perception might not be accurate. In our study, we found that students' self-assessment of learning using the perception questionnaire was in contrast with the assessment of learning via quizzes. Students' perceived that they had remembered better with games but we found no differences in quiz scores among lesson types. It was therefore important to verify self-reporting of perception with other modes of assessment such as behavioural observations and quizzes, as in our study.

Overall, we have found positive effects of EG over DI. In particular, there was an increased in interest in the topic, motivation to learn and attention retention. In addition, we were better able to engage the students, facilitate social learning and social bonding. The focus shifts from the teacher to the learners, acknowledging the students' voice as central to the learning experience. Today, information is a couple of clicks away and the education system should instead be focused on developing students' creative and communicative skills, motivation to succeed and ability to collaborate with others.

Educational reforms and pedagogical research are current hot topics (*Swallow, 2012*; *Wagner & Compton, 2015*). The aims are to shift from memory-based learning to meaning- and application-focused education, creating innovative thinkers and developing team-players. EG is one approach that could cater to these objectives and develop conservation biologists for the modern world.

To the best of our knowledge, this is the first study to examine the effects of games on conservation teaching. There is scope for many further studies. The effects of the EG approach could be tested on the general public which might have less interest for the work (in this case biodiversity conservation). We could examine if the EG approach is able to increase intrinsic motivation in non-conservationists, and cultivate people's minds to be more aware and involved in conservation issues. Further, it would be interesting to examine the effects of personality on group dynamics in an EG environment to better understand the interaction between personality traits and group performances (*Kramer, Bhave & Johnson, 2014*). Future work should also investigate the long-term effects of the training such as students' application of the acquired skills and friendship bonds.

## CONCLUSION

Experiential game provides a novel approach to conservation teaching. Despite the extra time needed for the tutor and for the lesson, our findings demonstrate the many benefits for the students as compared to DI. Conservation games, if used appropriately, can play an important role in making conservation more interesting and immersive. EG could also be an educational approach towards building capacity among conservationists in both NGOs and governmental organisations, and perhaps to engage multinational citizens and professionals with biodiversity conservation.

## ACKNOWLEDGEMENTS

We would like to thank the University of Nottingham Malaysia Campus for providing the ethics approval and facilities for the Wildlife Conservation Course. We would also like to thank the observers that have helped collected data on behaviours: ATM Perera, C Lim, A Lee, LY Chou, ABM Alias, WL Poh, MM Weerabangsa, MB Mahadzir, HK Wong, JY Ong and J Moore, the game designers E Arraut and J Lynch and teaching assistants J Wadey, A Tan, EP Wong and N Hii. A Hughes and L Porter.

### Funding

This work was supported by the Recanati-Kaplan Foundation and Panthera. The funders had no role in study design, data collection and analysis, decision to publish, or preparation of the manuscript.

### Grant Disclosures

The following grant information was disclosed by the authors:
Recanati-Kaplan Foundation and Panthera.

## Competing Interests

The authors declare that they have no competing interests.

## Author Contributions

- Cedric Kai Wei Tan conceived and designed the experiments, performed the experiments, analysed the data, contributed reagents/materials/analysis tools, prepared figures and/or tables, authored or reviewed drafts of the paper, approved the final draft.
- Jiin Woei Lee conceived and designed the experiments, performed the experiments, authored or reviewed drafts of the paper, approved the final draft, ethics clearance.
- Adeline Hii performed the experiments, analysed the data, prepared figures and/or tables, authored or reviewed drafts of the paper, approved the final draft.
- Yen Yi Loo performed the experiments, analysed the data, authored or reviewed drafts of the paper, approved the final draft.
- Ahimsa Campos-Arceiz conceived and designed the experiments, performed the experiments, contributed reagents/materials/analysis tools, authored or reviewed drafts of the paper, approved the final draft.
- David W. Macdonald conceived and designed the experiments, authored or reviewed drafts of the paper, approved the final draft.

## Human Ethics

The following information was supplied relating to ethical approvals (i.e., approving body and any reference numbers):

The University of Oxford Central University Research Ethics Committee granted Ethical Approval to carry out the study on study subject within the University of Oxford (Ref No: R42538/RE002).

The University of Nottingham Malaysia Campus Ethics Committee granted Ethical Approval to carry out the study on study subject within the University (ID #: CT011114, the initials are for the researcher Cedric Tan and the later numbers represent the date).

## Data Availability

Raw data is available at OSF: https://osf.io/tfuhg/.

## Supplemental Information

Supplemental information for this article can be found online at http://dx.doi.org/10.7717/peerj.4509#supplemental-information.

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
