# Peer review of "The effect of using games in teaching conservation"

_PeerJ, doi:10.7717/peerj.4509_

## Round 0.1 · original submission · Major Revisions

Please address all of the reviewers' comments, as listed below, In addition, please improve the following:

-Clarity of the protocol. It is quite difficult to follow the details of how the experiment was done.
-Demographics and number of participants used is unclear. Combine the participants details (main manuscript plus Suppl material) and write a clear paragraph, including a cohort demographics description table.
-Include the questionnaires and observations protocols used, or add citations for them at all times. In addition, discuss validation scores and explain why those surveys/observations protocols were chosen for this study. Discuss surveys and observation protocols limitations. Authors mention some modification to the surveys. More information and rationale is needed for these choices.
-Explain how observations were performed and discuss bias.
-The experimental design section does not fully describe the intervention and approach for teaching styles. Include a detailed explanation of the choices made for approaches, design and controls/experimental groups.
-Please discuss recent literature on learning styles and explain your experimental choice to use the 1994 survey.
-Please include Institutional review board information.

Importantly:
-Discuss how the study was framed to be objective towards the treatments.

Reviewer 1 ·

Basic reporting

The manuscript lacks clarity in many respects, and in particular on the methods and experimental design. The manuscript also needs additional review by an English speaker for clarity of language.

Experimental design

This study was designed to investigate the effect of using different types of games in conservation training courses for early-career professionals. The topic and questions explored in this study are interesting and worthy of attention, and the effort to collect data on instructional effectiveness and learning outcomes is commendable. The authors are clearly thoughtful educators and raise important questions in the discussion that are worthy of further investigation. However, the study seems to have a number of weaknesses in terms of the experimental design and data collection methods, and the manuscript lacks clarity in many respects. The latter would require significant re-writing, but the former are harder to address.

Validity of the findings

I was unable to fully evaluate the statistical analyses in detail because of my lack of familiarity with the methods used, but have important concerns about the methods.

Concerns with the experimental design and justification of methods:
- The overall design is not clearly described or justified. Table 1 supplies some information, but it is that comprehensive and leaves many open questions. How long was each game played for? Can they be considered analogous? How was the data binned or combined for analyses?
What would be required is a table or figure that clearly lays out: a global picture of how many students were included in the data collection in each course and each year, with each set of analogous questionnaires (since these changed over time), that is, which sets of student experienced the same questionnaires and design (repeated measures versus counterbalanced), and what the sample size was for each set.

- Why were two different designs used (repeated measures and counterbalanced)? There is no explanation for the choice of these approaches, nor sources cited, or why both were used in this study, nor how they affected the results. The potential for order effects is raised in the methods section (Line 178) but not furher discussed.

- The application of three types of instruction (supplememntal games, experiential games, and didactic instruction) is evaluated in terms of its effect on perception (through questionnaires), learning (through content quizzes), and behavior (through observations). Unsufficient information is provided on how these data collection tools were designed and validated and the questionnaires and quizzes themselves do not seem to be available. How were the questions formulated? Where they tested in any way? For example, in the observations of behavior, the asking/answering of questions was quantified - were the type of questions relevant? When observing and recording behaviors, was this done for the whole group playing the game, or were these recorded per individual? If a single individual asked most of the questions this could impact the data.

- Why were changes made to the questionnaires over time (as per lines 194-195) and how did these potentially affect the results? This is not explained or discussed.

- There is no discussion of the possible limitations of self-assessment questions.

A suggestion: A summary of good practices for validation can be found in Dirks, Clarissa, Mary Pat Wenderoth, and Michelle Withers. Assessment in the college science classroom. WH Freeman, 2014. Chapter 7.

Additional comments

Additional / specific comments:
Line 97 - I would disagree with the authors that conservation biology focuses on cause-and-effect systems. In my view, they are dynamic and complex systems with feedbacks and time-delays, not cause-and-effect. The authors should include citations to support this characterization.

- In Line 186 the authors state that additional assessment was done for a portion of the students after one week to assess "long-term” knowledge acquisition. I would disagree that one week can be considered long-term, or would like to see references supporting this characterization.

Line 205-207 is unclear.
Line 222 - it is not clear what is meant by “statement" here.

Lines 342-344 are unclear

Line 372 Sentence starting with "Second, instead of regurgitating..” is poorly written and unclear. It is unclear, who is regurgitating the knowledge here and what is
being proposed instead and on what grounds.

Line 391 - what is meant by “first-hand”?

Lines 354-356 The authors interpret the lack of interaction between personality scores and lesson type on questionnaire scores (behavior) as evidence that the preference for EG over DI is universal. I would argue that this but one interpretation. This could also be interpreted as limited sensitivity of these tools or the experimental design to changes in preferences. These other interpretations should also be discussed as opposed to making this strong claim.

Minor errors:
Line 366 centre or central?
Line 325 uleast = least
There is a typing error in Question 3 in Supplemental material “Learning_style_survey"

The manuscript needs additional review by an English speaker for clarity of language. For example, the second sentence in the abstract is confusing:
Line 22 - "Games are an increasingly popular approach for conservation teaching. However, we know little about the effectiveness of the games on conservation teaching.” Do the authors intend to mean effectivess of the teaching? That is not what is being measured in the study, but rather the effects of the games on student experiences and knowledge acquisition.

Other examples of sentences that would benefit from a review of language for clarity and precision are:
Line 24-25 - I think was is meant is: goal-oriented tasks found in real-life situations
Line 60 - both or all?
Line 95 emphasizes

Reviewer 2 ·

Basic reporting

The manuscript can be improved by making the writing clearer and more concise. Because there are many elements to the study, it is difficult to keep track of the different components, so making sure that the same terms are used, clarifying definitions, and putting details into table form can do much in helping the reader navigate. There also repetitions of statements and extraneous language, particularly in the introduction.

Experimental design

The authors have collected data on many aspects of the students' experience and performance over several groups and years, which makes it difficult to organize in a coherent manner. More clarity about some aspects of the methodology are needed, particularly why certain parts of the data collection only applied to certain groups and especially how biases were eliminated (example, the person conducting the sessions giving equal effort to all treatments; how the concepts and content of different modes of instruction were made comparable). The behavioural data collected on 'joyful behavior' seems particularly biased towards showing positive outcomes for the games versus the didactic instruction, so I am not sure if this should even be included. The presentation of the results may benefit from being summarized both in text and figures.

Validity of the findings

I do think the findings are interesting, but am concerned that there are statements that try to explicitly link what happens in the classroom setting will also result in behaviors and attitudes with positive outcomes for conservation are an overreach. I would suggest really being careful and specific about where the findings/conclusions of this study, especially outside of a learning environment, would apply. The conclusion section should probably be rewritten with this in mind.

Additional comments

I do think that this study has merit and interesting results, but how the manuscript was put together was confusing at times. I think that it can be made more coherent and cohesive - links are needed between the perception of the students about the experience of games in the classroom and the benefits this provides for learning or social outcomes. I provide more comments in the pdf which I hope will improve it.

Annotated reviews are not available for download in order to protect the identity of reviewers who chose to remain anonymous.

Reviewer 3 ·

Basic reporting

High standard of academic English used that was clear and easy to understand to someone versed in the discipline.
Good and correct use of referencing.
Some of the data could perhaps be more easily conveyed to the reader through the used of tables or charts.
Overall well structured and presented.

Experimental design

Overall a very good experimental design that used multiple approaches to triangulate th results of the data.
Some more description of the observational aspects of the research (e.g. observing the behaviour students) might be required within the main text of the article.

Validity of the findings

Findings appear valid and well constructed.
The use of multiple avenues of data collection as well as more than one research design (i.e. both counterbalanced and repeated-measures) provided reliability to the data.

Additional comments

Overall an excellent paper. Some minor adjustments in terms of the presentation of the results and the data could just improve the overall quality of the paper.

---

## Round 0.2 · Minor Revisions

Let me start by saying that we truly appreciate the expensive revisions you have introduced. It has dramatically improved the manuscript.
Before the current version is ready for publication, please address the following:

-Despite the popularity of learning styles there is no evidence to support the idea that matching activities to one’s learning style improves learning. Please discuss the work of Harold Pashler, Mark McDaniel, Doug Rohrer, and Robert Bjork. The instrument and learning styles you are using doesn't seem to be up to date with current literature in the field. Please discuss this.

-In assessments you describe quizzes pre and post-session to examine the knowledge acquired by each individual. Where the questions administered the same pre and post? if not, how where they compared pre/post? I understand they were classified in different types (reproduction-, meaning- or application) complementary to the learning styles, can you clarify what this means? How do they classify in terms of cognitive level/difficulty (e.g. Blooms taxonomy)? and how are they matched pre and post? Are the questions validated (for difficulty, discrimination and reliability)? Psychometric item validation is recommended.

-In assessments you also mention that behavior was evaluated via observational surveys and video recording during the classroom. Can you add citations and/or describe the validity of protocols? if the protocol is new, we suggest adding statistical validation of the method used.

-Figure 1 doesn't explain how supplemental game or experiential game were alternated. The experimental design should be reflected in every figure. For example, how do results in figure 2 aligning with design in figure 1. Clarify.

-Figure 2: please clarify how each score shown in figure 2 was calculated. Clarify meaning of axes. e.g. "after minus before relationship"?

-Figure 4 must include description and validation of items used to calculate learning pre and post. Are questions used the same? comparing apples to apples? or difficulty level (e.g. Blooms taxonomy) varies?

-All supplementary material figure legends need to include correlation coefficients, statistics associated to it. For tables the variables tested are hard to follow.

-All legends need to be expanded to clarify variables, calculated scores, validation and statistics used.

---

## Round 0.3 · accepted · Accept

Thanks for addressing the comments I made. The manuscript is now ready for publication, congratulations!